# Effects of At-Home and In-Office Bleaching Agents on the Color Recovery of Esthetic CAD-CAM Restorations after Red Wine Immersion

**DOI:** 10.3390/polym14183891

**Published:** 2022-09-17

**Authors:** Wei-Fang Lee, Hidekazu Takahashi, Shiun-Yi Huang, Jia-Zhen Zhang, Nai-Chia Teng, Pei-Wen Peng

**Affiliations:** 1School of Dentistry, College of Oral Medicine, Taipei Medical University, Taipei 11031, Taiwan; 2School of Dental Technology, College of Oral Medicine, Taipei Medical University, Taipei 11031, Taiwan; 3Course for Oral Engineering, Faculty of Dentistry, Tokyo Medical and Dental University, 1-5-45, Yushima, Bunkyo-ku, Tokyo 113-8549, Japan

**Keywords:** bleaching, esthetic CAD/CAM, color, whiteness index, red wine immersion

## Abstract

The aim of this study was to evaluate the effects of at-home and in-office bleaching agents on esthetic CAD-CAM materials after red wine immersion by measuring their optical properties. Sixty specimens were prepared out of three esthetic CAD-CAM materials: Vita Enamic, Celtra Duo, and Ceresmart (*n* = 20). All specimens were immersed in a red wine solution, and color measurements were performed. Specimens were randomly divided (*n* = 10) according to the bleaching procedure (in office, at home), bleaching durations were set to 3 time points, and color measurements were performed. According to the Commission Internationale de l’Eclairage (CIE) *L** *a** *b** parameters, CIEDE2000 color differences (Δ*E*_00_), translucency parameters (TP_00_), and whiteness index values (ΔWI_D_) after wine staining and after bleaching were calculated. Data were analyzed using the Mann–Whitney U-test, the Kruskal–Wallis test, and a two-way analysis of variance (ANOVA) (α = 0.05). Δ*E*_00_, ΔTP_00_, and ΔWI_D_ decreased with an increase in bleaching treatment. Δ*E*_00_ after the final bleaching treatment of in-office bleaching ranged from 1.7 to 2.0, whereas those of in-office treatment ranged from 0.4 to 1.1. All ΔTP_00_ and ΔWI_D_ after the final treatment were below the 50:50% perceptibility thresholds (ΔTP_00_ < 0.6, and ΔWI_D_ < 0.7). Significant differences in Δ*E*_00_, ΔTP_00_, and ΔWI_D_ among esthetic CAD-CAM materials were found between CD and CE. In the present study, color recovery after at-home and in-office bleaching appeared to be material-dependent. In-office bleaching showed more effective recovery comparing to at-home bleaching.

## 1. Introduction

Nowadays, aesthetic appearance in contemporary dentistry is defined as a natural, beautiful, and confident smile, so bleaching and smile design have become popular in aesthetic dentistry [1,2]. Both intrinsic and extrinsic factors often cause discoloration of restorations. Factors that influence the formation of extrinsic stains include poor oral hygiene, smoking, and food colorants such as red wine, tea, and coffee, whereas the intrinsic stains may result from an alteration in the tooth material (e.g., oxidation of residual monomers) [3,4,5,6,7].

Tooth whitening has become a popular procedure in cosmetic dentistry and is an effective and relatively safe esthetic treatment [8,9]. There are various tooth-bleaching products on the market, which are clinically divided into two types [10]. In-office bleaching, commonly performed with a high concentration of hydrogen peroxide (HP) or carbamide peroxide (CP) for 15~60 min, is widely applied because of the benefits such as a rapid response and the protection of soft tissues [11]. For at-home bleaching, patients may use a lower concentration of CP or HP for 1~4 weeks in an easy-to-use process [10,12]. Bleaching agents were an effective method of stain removal and color recovery [13]. Previous research showed that tooth whitening significantly enhances people’s self-confidence [14], and with an increasing desire for white teeth, the demand for tooth whitening and the use of tooth-bleaching products have increased [8,9].

Color alteration of the restoration can be detected using a spectrophotometer, which records three color parameters [15]. According to the Commission Internationale d’Eclairage *L** *a** *b** (CIELab) color system, *L** indicates darkness to lightness, the *a** coordinate represents the green to red range, and the *b** coordinate represents the blue to yellow range [12]. Based on the CIEDE 2000 system, the 50:50% perceptible (PT) and acceptable (AT) thresholds are determined as 0.8 and 1.8. The corresponding PT and AT of whiteness index (WI_D_) are considered to be 0.72 and 2.60, respectively [12,16,17].

Dental resin composites containing polymer and salinized organic filler were susceptible to staining [16]. A combination of polymer, ceramic, and computer-aided design (CAD) and computer-aided manufacturing (CAM) technologies have been developed to overcome drawbacks such as polymerization shrinkage, monomer release, and a lack of color stability of these materials. A wide range of CAD-CAM blocks, especially regarding esthetic restorative materials, have been developed, including composite resins, polymer-based composite, polymer-infiltrated ceramics (hybrid ceramics), and zirconia-reinforced lithium silicate (ZLS) [17,18]. These CAD-CAM materials are less brittle, have lower stiffness and hardness, can be more easily machined, and are more tooth-friendly [19,20].

While keeping color stability of dental restorations in an oral environment is a challenging yet essential element in producing successful restorations [21]. The adsorption of the colorants from the external staining, especially red wine, was one of the main factors causing discolorations of the esthetic CAD-CAM materials [22,23]. Regarding discoloration of the esthetic CAD-CAM blocks, color changes after 28-day immersion in red wine were significantly different in the following order: composite > hybrid ceramics > lithium disilicate [24]. Aydın et al. [14] investigated the effects of different beverages on the color changes of the esthetic CAD-CAM blocks and concluded that all examined materials, including hybrid ceramics, composite, and zirconia-reinforced lithium silicate, showed clinically unacceptable color differences after 30-day immersion in red wine and coffee.

Discolorations could be removed using different strategies [24,25,26]. The bleaching agents with different concentrations of HP are effective for removing stains. However, less information is available concerning the color recovery of red-wine-immersed CAD-CAM materials. Thus, the present study aimed to evaluate the effects of two different bleaching agents on restoring the color before and after accelerated staining and on the optical properties of the esthetic CAD-CAM materials, including two hybrid ceramics and ZLS. The null hypothesis was that the recovery of the optical properties after red wine staining would not be influenced by the bleaching agents and esthetic CAD-CAM materials.

## 2. Materials and Methods

Information on three different esthetic CAD-CAM blocks, one staining solution, and two types of bleaching agents used in the present study is given in Table 1. All specimens were used according to the respective manufacturer’s instructions.

### 2.1. Specimen Preparation

According to previous studies, the specimen for the optical measurements was prepared in a square shape with the dimension of 10 × 10 × 2 mm^3^ [27]. Twenty specimens for each CAD-CAM material were prepared using a cutting machine with a diamond saw (series 15LCU, Buehler, Lake Bluff, IL, USA) and finished with 1200 grit SiC paper. For the CD group, specimens were fired in a ceramic furnace (Programat P700; Ivoclar Vivadent AG, Schaan, Liechtenstein) at 820 °C for 8 min. After rinsing and gentle drying, the thickness of the specimens was determined with a digital micrometer (MDC-250; Mitutoyo, Kawasaki, Japan). All specimens were cleaned with distilled water in an ultrasonic cleaner for 5 min, dried, and stored in distilled water at 37 °C for 24 h, and color measurement was performed and recorded as the time point of the baseline (denoted as T_B_ and T_OB_).

### 2.2. Color Measurements

The color of each specimen was measured at different measuring times: before (T_AB_, T_OB_) and after staining with red wine (T_AS_, T_OS_), with at-home bleaching at 2 (T_A1_), 4 (T_A2_), and 8 days (T_A3_), and with in-office bleaching at 7.5 (T_O1_), 15 (T_O2_), and 22.5 min (T_O3_).

The color of each specimen was measured using a dental spectrophotometer (ShadePilot; Degudent, Rodenbacher, Hanau-Wolfgang, Germany) on standard white (*L** = 92.28, *a** = −1.28, and *b** = −2.05) and black (*L** = 1.50, *a** = −2.37, and *b** = −8.41) backgrounds. The spectrophotometer was calibrated using a white tile before each measurement, and the center of each specimen was measured three times. The means and standard deviations (SDs) of the *L**, *a**, and *b** parameters were recorded.

Compared to the color against a black background at different times, the color difference in the specimen at different bleaching times was quantitatively calculated using the CIE2000 formula (Δ*E*_00_) [28,29,30]:(1)$$\Delta E_{00}=\sqrt{{\left(\frac{L_{i}-L_{j}}{K_{L}S_{L}}\right)}^{2}+{\left(\frac{C_{i}-C_{j}}{K_{C}S_{C}}\right)}^{2}+{\left(\frac{H_{i}-H_{j}}{K_{H}S_{H}}\right)}^{2}+R_{T}\left(\frac{C_{i}-C_{j}}{K_{C}S_{C}}\right)\left(\frac{H_{i}-H_{j}}{K_{H}S_{H}}\right)},$$
where C refers to chroma, H refers to hue, the subscripts i and j refer to values obtained from different periods, S_L_, S_C_, and S_H_ represent weighting functions, K_L_, K_C_, and K_H_ are parametric factors, which were set to 1 in the present study, and RT is the rotation function.

Similarly, the translucency parameter (TP_00_) was calculated as follows [31,32]:(2)$${\mathrm{TP}}_{00}=\sqrt{{\left(\frac{L_{B}-L_{W}}{K_{L}S_{L}}\right)}^{2}+{\left(\frac{C_{B}-C_{W}}{K_{C}S_{C}}\right)}^{2}+{\left(\frac{H_{B}-H_{W}}{K_{H}S_{H}}\right)}^{2}+R_{T}\left(\frac{C_{B}-C_{W}}{K_{C}S_{C}}\right)\left(\frac{H_{B}-H_{W}}{K_{H}S_{H}}\right)},$$
where B and W respectively refer to a specimen placed on a black and white background, respectively.

The whiteness index for dentistry (WI_D_) was calculated according to the following equation [28,33,34]:(3)$${\mathrm{WI}}_{D}=0.511L^{*}-2.324a^{*}-1.110b^{*},$$

High positive values of the difference of WI_D_ between two specimens (ΔWI_D_) indicate higher whiteness values of specimens.

### 2.3. Staining Procedure

The staining procedure was described detail in a previous study [26]. Briefly, all specimens were immersed in a red wine solution at room temperature. The solution was refreshed daily to avoid bacterial and fungal contamination. All specimens were immersed in a red wine solution at room temperature for 7 days. The solution was refreshed daily to avoid bacterial and fungal contamination. The color of stained specimens was measured and recorded as the time point of staining (denoted as T_AS_ and T_OB_). After staining, each specimen was washed and stored in 37 °C distilled water until the bleaching procedures.

### 2.4. Bleaching Procedures

After cleaning in distilled water for 5 min, the specimens in each material group were randomly divided into two groups (*n* = 10), according to different bleaching treatments.

#### 2.4.1. At-Home Bleaching Procedures

An at-home bleaching agent containing the 16% HP was applied to the surface of each specimen twice daily for 8 days according to the manufacturer’s instructions. A bleaching agent of 2 mL was applied on the top surface of the specimen with a duration of 3 h, and then the specimen was rinsed with distilled water for 1 min and dried with tissue paper. Two sets of this process were repeated every day. Color measurements were taken at 2, 4, and 8 days. For easy identification, they were denoted as T_A1_, T_A2_ and T_A3_.

#### 2.4.2. In-Office Bleaching Procedures

An in-office bleaching gel containing 40% HP was used. For each application, the specimen was coated with 2 mL bleaching agent to be a uniform layer of ~1 mm in thickness and activated for 7.5 min using an LED light-curing device (3M ESPE Dental Products, 3M Oral Care, Monrovia, CA, USA) in the plasma emulation mode with an intensity of 3200 mW/cm^2^. The specimen was then washed with distilled water and gently dried with tissue paper. Subsequently, another 5 mL of bleaching agent was applied, and the application was repeated three times. For easy identification, they were denoted as T_O1_, T_O2_, and T_O3_.

### 2.5. Statistical Analysis

All statistical analyses were performed using statistical software (SPSS Statistics v.26, IBM, Chicago, IL, USA). Descriptive statistics are shown as the mean ± standard deviation. The normality of the distribution was analyzed using Shapiro–Wilk test. The Mann–Whitney U-test and Kruskal–Wallis test were used as post hoc tests for statistically significant variables. Results were interpreted using the Bonferroni correction (*a* = 0.05). A two-way analysis of variance (ANOVA) was used to analyze the influence of material types, bleaching methods, and the interaction of these on Δ*E*_00_, ΔTP_00_, and ΔWI_D_ values.

## 3. Results

The immersion period of all materials was 7 days when the Δ*E*_00_ between T_B_ and T_S_ was greater than 6. The mean Δ*E*_00_ between after staining (T_AB_ or T_OB_) and before staining (T_AB_ or T_OB_) and those among different periods of bleaching applications are presented in Figure 1.

Results of the two-way ANOVA for optical parameters of all groups are shown in Table 2. The Δ*E*_00_ and ΔTP_00_ demonstrated that main factors of the type of material and bleaching agents and their interaction were significant (*p* < 0.05). For ΔWI, there was no significant interaction between materials and bleaching agents (*p* = 0.137).

Results of ΔTP_00_ values are illustrated in Figure 2. The ΔTP_00_ values after staining, the significant difference between VE and CD of at-home bleaching (*p* < 0.001) and the difference between CD and VE (*p* < 0.037) and between CD and CE (*p* < 0.006) of in-office bleaching were detected. Among the bleaching measuring time points of each material, there was no statistically significant difference. Comparing ΔTP_00_ of three materials at the same bleaching application, no significant difference was detected except for VE and CD of at-home bleaching of T_A2_-T_AS_ and T_A3_-T_AS_, which was a similar tendency of Δ*E*_00_.

The results of ΔWI_D_ values are presented in Figure 3. The ΔWI_D_ values after staining and the significant difference between VE and CD of at-home bleaching (*p* < 0.002) was detected. For at-home bleaching at all bleaching measuring time points, there were significant differences between VE and CD at T_A1_-T_AS_ (*p* < 0.01), T_A2_-T_AS_ (*p* < 0.025), and T_A3_-T_AS_. (_S_ < 0.02). For in-office bleaching at all bleaching measuring time points, there were no significant differences.

Comparing among Δ*E*_00_ values of T_B_-T_S_, the Δ*E*_00_ values of at-home bleaching of VE and CD was significantly different (*p* < 0.002), but those of in-office bleaching were not significantly different. After the first bleaching of T_A1_ and T_O1_, the Δ*E*_00_s for each material of T_A1_-T_AS_ and T_O1_-T_OS_ were not significantly different from those of T_AB_-T_AS_ and T_OB_-T_OS_, respectively. There were no significant differences among the Δ*E*_00_s of T_A1_-T_AS_, T_A2_-T_AS_ and T_A3_-T_AS_ or among Δ*E*_00_s of T_O1_-T_OS_, T_O2_-T_OS_ and T_O3_-T_OS_ for each material. Comparing the Δ*E*_00_ values of three materials at the same bleaching application, no significant difference was detected except in the VE and CD of at-home bleaching of T_A1_-T_AS_ and T_A2_-T_AS_. The Δ*E*_00_s between the baseline (T_AB_, T_OB_) and different periods of beaching applications of each material of at-home bleaching and in-office bleaching are illustrated in Figure 4a and b, respectively. The ΔTP_00_s among the baseline and different periods of beaching application of each material are illustrated in Figure 4c,d. The effects of multi-application of bleaching and bleaching material showed a similar tendency of ΔTP_00_. All ΔTP_00_s after T_A3_ and T_O3_ were within the in vitro PT of 0.6 [35]. The ΔWIs among the base line and different periods of beaching application of each material are illustrated in Figure 4e,f. Effects of multi-applications of bleaching and bleaching materials showed a similar tendency of ΔWI. All ΔWIs after bleaching were within the in vitro 50:50% acceptability threshold (AT) of 2.6 [35].

The changes in *L**, *a**, and *b** between the baseline, after staining, and after the bleaching treatments are shown in Figure 5. Generally, *L** and *b** decreased and *a** increased after red wine staining, which means samples became dark, blueish, and reddish, but *L**, *a**, and *b** after bleaching almost recovered from staining to the baseline level.

## 4. Discussion

The present study evaluated the effect of two different bleaching agents on esthetic CAD/CAM materials after red wine staining. The null hypotheses were that the optical property recovery after red wine staining would not be influenced by the bleaching agents and esthetic CAD-CAM materials. According to the results obtained the present study, there were significant differences in color, translucency, or whiteness variations by bleaching agent and materials. Therefore, the first hypothesis was rejected.

For the optical measurements, the thickness of specimens was varied from 0.5 mm to 2 mm [27]. A 2 mm thickness used in the present study was considered to reduce the effect of the background. Few studies have investigated accelerated staining and then used bleaching agents to observe whether the original color of the restoration could be restored. Red wine contains phenolic compounds such as tannins and anthocyanins, and is a beverage most likely to cause staining [6]. According to previous reports, it is known that because red wine contains alcohol, a complex process of surface deposition causes degradation of the resin matrix by ethanol and the acid of pigments, which may further induce discoloration by increasing the adsorption of pigments in the surface of a restoration [3,4,6]. Therefore, red wine was used as the staining solution for its effect on the color stability of esthetic CAD-CAM restorations in the present study.

CAD-CAM esthetic blocks are a widely used material for dental esthetic restorations, and due to esthetics demands, bleaching treatments have become popular and can be performed by patients using routine procedures. Therefore, the pursuit of esthetic restorations to maintain color and translucency stability is an important factor in the success of prosthetic restorations.

Paravina et al. indicated that the CIEDE2000 formula better reflects the human perception of color differences than the CIELAB formula, and it can improve the perceptibility and acceptability of color differences in oral conditions [13,28]. All Δ*E*_00_, ΔTP_00_, and ΔWI_D_ in the present study after finishing bleaching treatments were below the ATs (Δ*E*_00_ < 1.8, ΔTP_00_ < 2.6, and ΔWI_D_ < 2.6) except for the Δ*E*_00_ of at-home bleaching. Bleaching treatments, including both at-home and in-office bleaching agents, were demonstrated to be effective in improving stain removal for resin-based materials, but they caused color changes [2]. A previous study using 15% CP (8 h/day for 4 weeks) resulted in a clinically acceptable visual threshold (Δ*E***ab* < 2.72) for the resin composite [36]. However, after application of 10% CP (8 h each time for 14 days) or 20% CP (6 h/day for 8 days), the color change of the resin composite was not clinically detectable [31,33]. Kim et al. also reported that CP did not cause perceptible color change in nano-filled or micro-hybrid resin composites [29]. Our results demonstrated that even 15% CP can restore a clinically acceptable visual threshold. Therefore, it was reported that the same tooth color change may be produced using a low-concentration H_2_O_2_ bleaching agent with the advantage of lower risk and sensitivity [9]. Hydrogen peroxide is an aggressive oxidant that causes unpolymerized monomers and non-specific oxidation products to elute from composites [36]. A previous study pointed out interactions between nano-hybrid materials with different degrees of polymerization and the effect of bleaching agents [37], and this needs to be confirmed in further study. Limitations of the present study include the fact that intraoral conditions were not replicated, so the use of artificial saliva and thermal cycling should be further simulated.

Regarding discoloration after 28-day red wine immersion, VE and CE were significantly higher than CD, which agreed with the previous report [26]. These differences were highly related to the different composition of materials. CD comprised crystalline minerals and a glass matrix, which has a dense microstructure, inhibiting the penetration of the staining solution. It could be deduced that discolorations of CD could almost be completely removed by the external modalities, such as bleaching treatment.

Differences between the polymer-infiltrated ceramics and polymer-based composite observed in this study may be related to the production technology of the materials, despite the similarity in the organic structure of the groups. In the present study, only the Δ*E*_00_ of all at-home bleaching and CE of in-office bleaching and ΔWI_D_ of VE and CE at-home bleaching and CE of in-office bleaching were greater than PT (Δ*E*_00_ < 0.8, ΔTP_00_ < 0.6, and ΔWI_D_ < 0.7) (Figure 4). Although there were some changes in the detection of optical properties of the Vita Enamic (polymer infiltrated ceramics) and Ceresmart (polymer-based nano composite) materials in the at-home bleaching group, particularly in the results of Δ*E*_00_ values, these color alterations would be detectable by standard observers. These findings are also consistent with our results that the color changed after accelerated staining (Δ*E*_00_: 11.73 to 13.53).

Translucency is an important factor for satisfying aesthetic properties, and previous studies indicated that bleaching increases the surface roughness and reduces the translucency of polymer materials [38]. They are affected by a difference in the refractive index between the filler and resin matrix, filler size, and fraction [13]. Moreover, the opaque esthetic blocks exhibited better color stability when immersed in red wine, which also validated our findings that the CD group had the least change in ΔTP_00_ because CD is a resin-free zirconia-reinforced lithium silicate ceramic. Δ*E*_00_ and ΔWI_D_ values of CD showed minimal changes, which was consistent with previous studies [14,32]. On the other hand, the whitening effect provided sufficient information about whiter or darker changes in a specimen after bleaching [39,40], and the present study used the ΔWI to validate the results. The results showed ΔWI_D_ values in the CD and CE specimens, which were brighter than the baseline (Figure 4), which is consistent with previous findings [41,42].

## 5. Conclusions

Within the limitations of this laboratory study, it was found that color recovery of the esthetic CAD-CAM materials after red wine staining by using at-home and in-office bleaching appeared to be material-dependent. In-office bleaching showed more effective recovery compared to at-home bleaching, especially VE and CD. For each material, the Δ*E*_00_ values in the in-office group were smaller than the 50:50% acceptability threshold after the final bleaching treatment, whereas those in the at-home bleaching group were still larger than 50:50% acceptability threshold values.

## Figures and Tables

**Figure 1 polymers-14-03891-f001:**
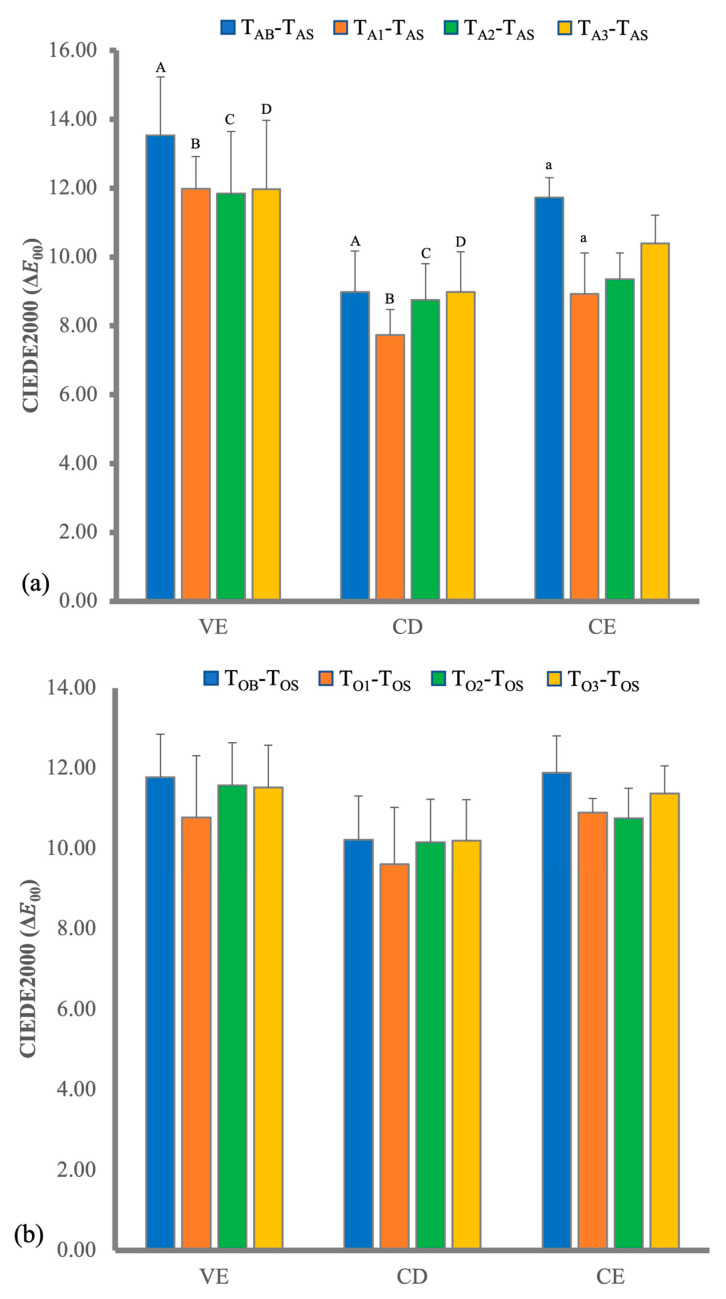
CIEDE2000 values (Δ*E*_00_) of the three materials at different periods for (**a**) at-home bleaching, and (**b**) in-office bleaching. VE, Vita Enamic; CD, Celtra Duo; CE, Ceresmart. Subgroups identified by same superscript lowercase letters were significantly different within groups; subgroups identified by same superscript uppercase letters were also significantly different between groups (*p* < 0.05).

**Figure 2 polymers-14-03891-f002:**
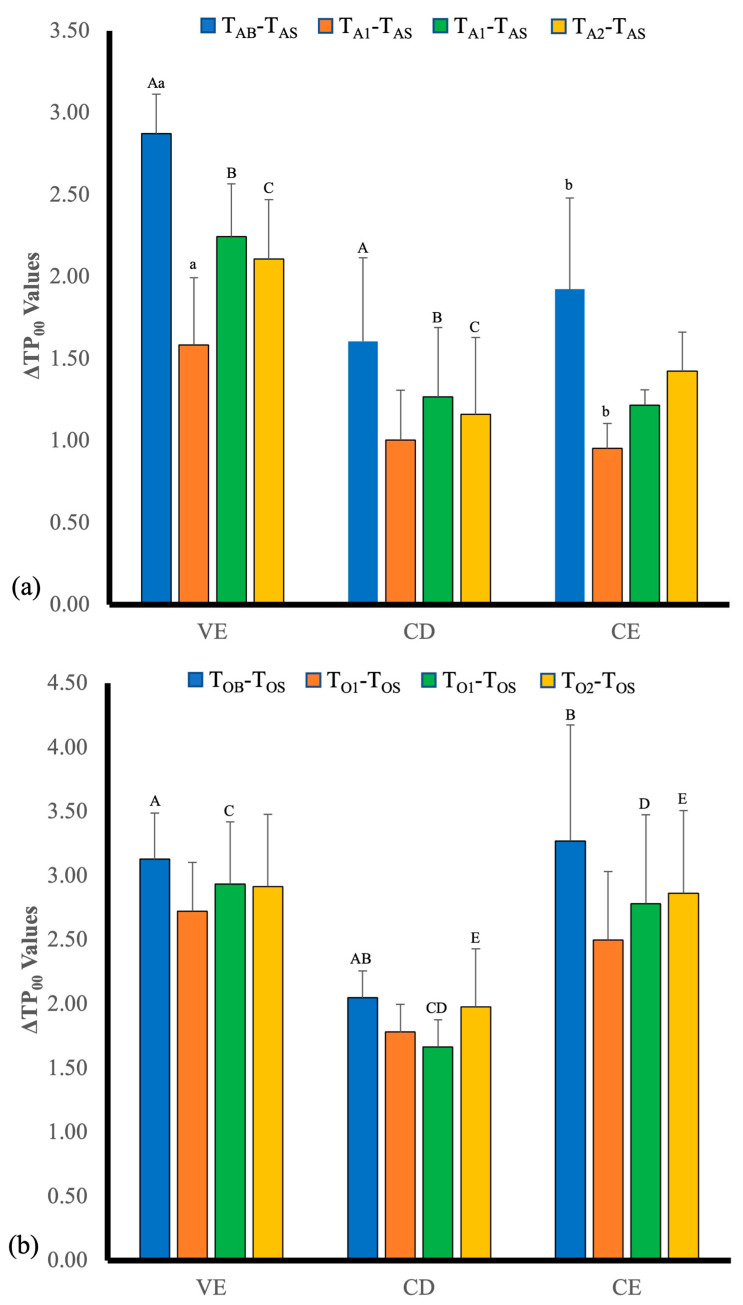
ΔTP_00_ values of the three materials at different times for (**a**) at-home bleaching, and (**b**) in-office bleaching. VE, Vita Enamic; CD, Celtra^®^ Duo; CE, Ceresmart. Subgroups identified by same superscript lowercase letters statistically differed within groups; subgroups identified by same superscript uppercase letters also statistically differed between groups (*p* < 0.05).

**Figure 3 polymers-14-03891-f003:**
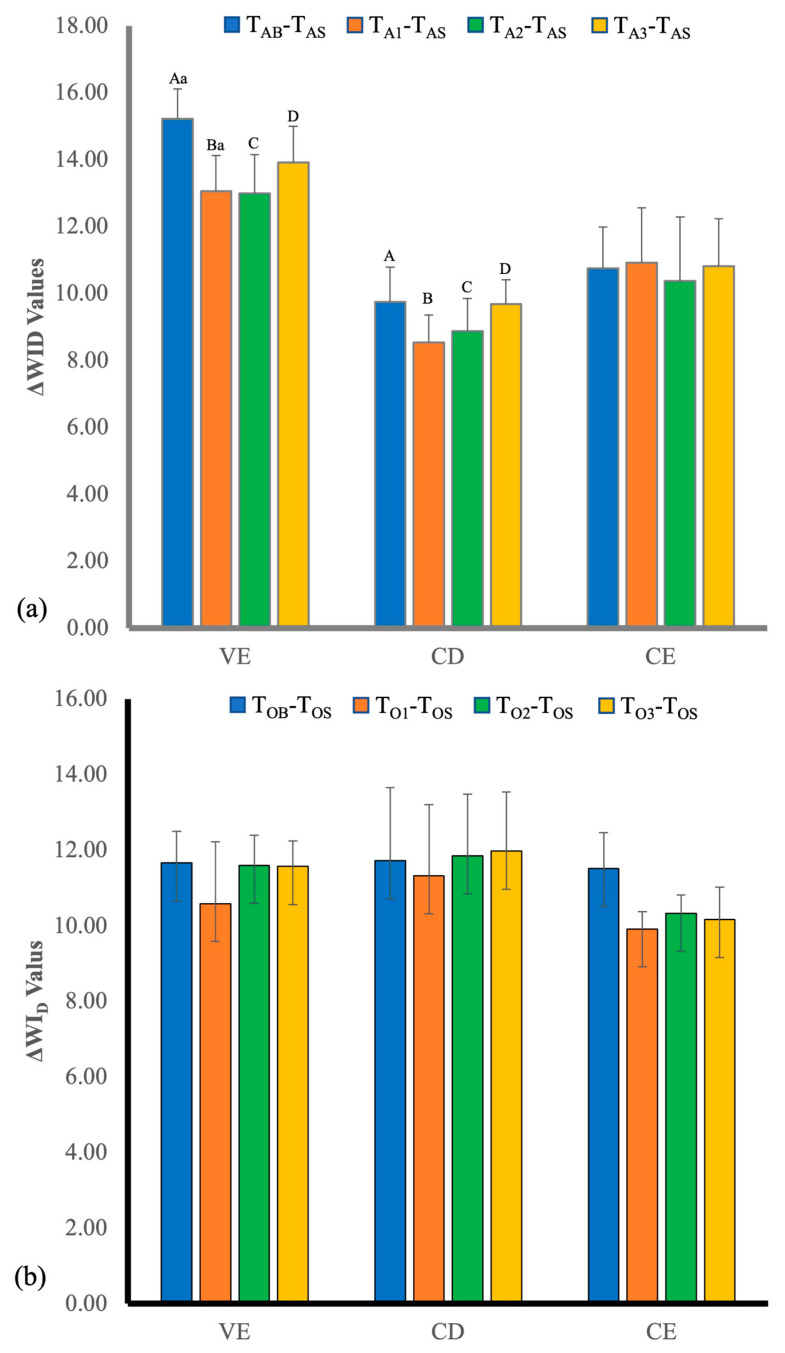
ΔWI_D_ values of the three materials at different times for (**a**) at-home bleaching, and (**b**) in-office bleaching. VE, Vita Enamic; CD, Celtra^®^ Duo; CE, Ceresmart Subgroups identified by same superscript lowercase letters statistically differed within groups; subgroups identified by same superscript uppercase letters also statistically differed between groups (*p* < 0.05).

**Figure 4 polymers-14-03891-f004:**
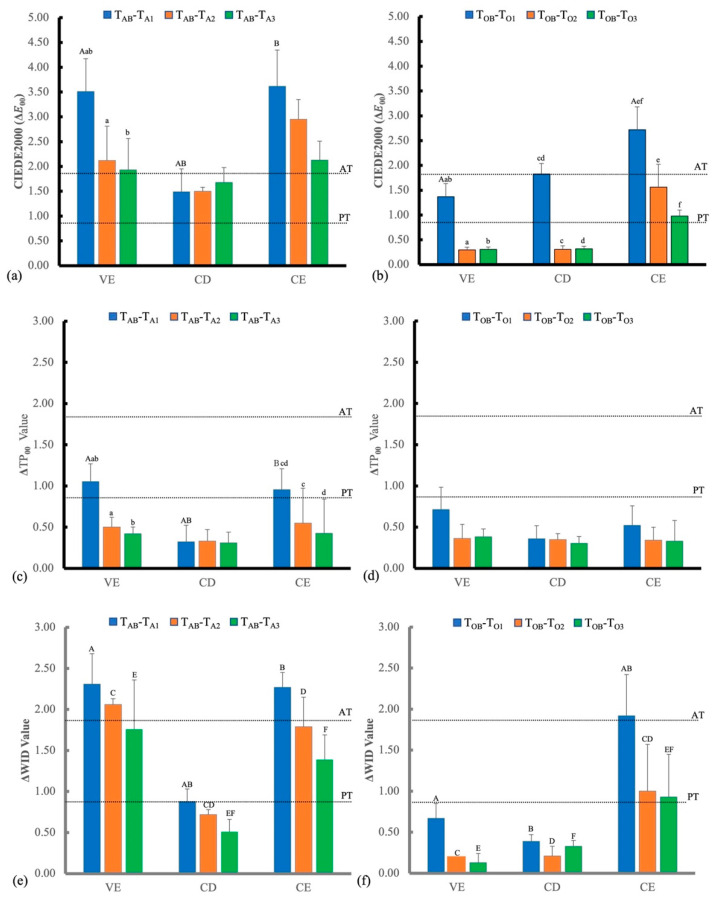
Color difference (Δ*E*_00_), changes in translucency parameter (ΔTP_00_), and changes in whiteness index (ΔWI_D_) values of the three materials before (T_AB_, T_OB_) and at different bleaching times for each material from the at-home (**a**,**c**,**e**) and in-office (**b**,**d**,**f**) groups. The 50:50% perceptibility threshold (PT) for Δ*E*_00_, ΔTP_00_, and ΔWI were 0.8, 0.6, and 0.7, respectively, and the 50:50% acceptability threshold (AT) for Δ*E*_00_, ΔTP_00_, and ΔWI were 1.8, 2.6, and 2.6, respectively. Subgroups identified by same superscript lowercase letters statistically differed within groups; subgroups identified by same superscript uppercase letters also statistically differed between groups (*p* < 0.05).

**Figure 5 polymers-14-03891-f005:**
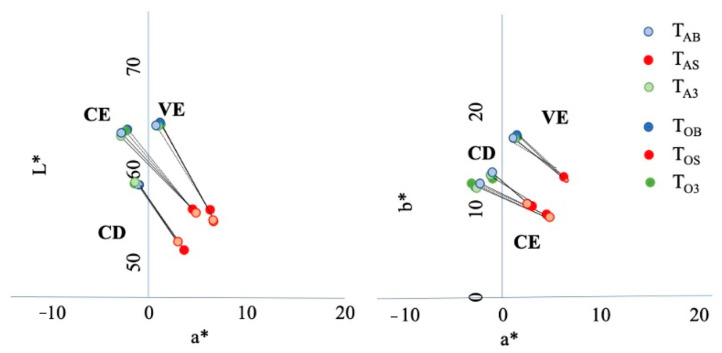
The changes in *L**, *a**, and *b** between base line to staining those between staining and bleaching.

**Table 1 polymers-14-03891-t001:** Materials and bleaching agents used in the present study.

Material	Code	Type	Shade	Manufacturer	Composition and Structure	Batch
CAD-CAM blocks						
Vita Enamic	VE	polymer-infiltrated ceramics	3M2-H2	Vita Zahnfabrik; Bad Sackingen, Germany	86 wt% feldsparthic-based ceramic (SiO_2_, Al_2_O_3_), 14% acrylate polymer (UDMA, TEGDMA)	54,073
Celtra Duo	CD	zirconia-reinforced lithium silicate	A3-LT	Dentsply Sirona, Bensheim, Germany	58 wt% SiO_2_,18 wt% LiO, 10.1 wt% ZrO_2_, 5 wt% Phosphorus pentoxide, 1.9 wt% Al_2_O_3_, etc.	18,018,969
Ceresmart	CE	polymer-based composite	A3-HT	GC Dental ProducTS; Europe, Leuven, Belgium	80 wt% nanoceramic fillers (SiO_2_ and barium glass), 20 wt% Acrylate polymer (Bis-MEPP, UDMA, DMA)	1,611,281
Bleaching agent						
Flash Take Home	at-home	Home bleaching agent		Whitesmile GmbH, Germany	16% carbamide peroxide, 5.6% hydrogen peroxide	1,903,017
Power Whitening	in-office	In-office bleaching agent		Whitesmile GmbH, Germany	40% carbamide peroxide, 32% hydrogen peroxide	1,903,017
Staining solution						
Red wine		Cabernet Sauvignon		Casillero del Diablo, Chile		2020

**Table 2 polymers-14-03891-t002:** Results of a two-way ANOVA for Δ*E*_00_, ΔTP_00_, and ΔWI_D_ values of all groups.

Value	Effect	Type III Sum of Squares	df	Mean Square	F	*p* Value
Δ*E*_00_	Bleaching	3.596	1	3.596	80.928	0.000
	Materials	5.134	2	2.567	57.773	0.000
	Bleaching × Materials	1.801	2	0.901	20.270	0.000
ΔTP_00_	Bleaching	3.690	1	3.690	16.228	0.003
	Materials	1.599	2	0.799	3.515	0.000
	Bleaching × Materials	0.906	2	0.453	1.991	0.001
ΔWI_D_	Bleaching	2.884	1	2.884	9.909	0.006
	Materials	5.866	2	2.933	10.076	0.001
	Bleaching × Materials	1.293	2	0.646	2.220	0.137

Δ*E*_00_, color difference; ΔTP_00_, difference in the translucency parameter; ΔWI, difference in the whiteness index value; df, degrees of freedom.

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
