# Peer review of "Effects of At-Home and In-Office Bleaching Agents on the Color Recovery of Esthetic CAD-CAM Restorations after Red Wine Immersion"

_polymers, 2022, doi:10.3390/polym14183891_

Round 1

Reviewer 1 Report

Dear author

Please see my suggestions below

In Table 1 please mention lot number for materials

Please use editor equation from Microsoft office!!! Please mention number of equation.

Figure 1. plese include the down in the graphs Please use for all graphs

Figure 4. is not so visible. Why some of graphs are colored and some of them not. Pleas made for all

Please prepare conclusion again and avoid numbers.

Author Response

Dear reviewer:

Please find the revised paper entitled “Effects of at-home and in-office bleaching agents on the color recovery of esthetic CAD-CAM restorations after red wine immersion” (Manuscript ID: polymers-1869164) enclosed. We have incorporated all the suggestions made by the reviewers. These changes are highlighted in red within the manuscript. Please see below for a point-by-point response to each reviewer’s comments and concerns.

Reviewer 2 Report

I’ve reviewed the manuscript (polymers-1869164) titled ‘Effects of at-home and in-office bleaching agents on the color recovery of esthetic CAD-CAM restorations after red wine immersion’ with interest and following are my comments and recommendations. 

1.     Introduction: A brief overview of the relevant literature related to colour stability, bleaching after red wine is missing. It would be good to see some previous data/recommendation from previous studies here especially related to the materials used in the present study.

2.     A comparison of hybrid CAM ceramic, resin composite and CAD-CAM ceramic with regards to colour stability would be good idea.

3.     Table 1, there is no mention of staining solution (red wine)’ as mentioned in Line 88, please provide details.

4.     Please provide a reference and justifications for the selection of bar-shaped specimen of dimension 10x10x2 mm3 preparation.

5.     Line 130, ‘Stained specimens from a previous investigation (reference number 30) were used in this study’, I didn’t understand this rationale. Didn’t you perform the staining procedure for this study particular? Please elaborate.

6.     Line 134, what do you mean by ‘stained specimens stopped immersing until the CIEDE2000 color difference was larger than 6….’, was there any standard storage duration, how would you possibly extrapolate this to a clinical situation?

7.     Line 144, ‘A bleaching agent of 2 ml was applied on the top surface of the specimen with a duration of 3 hours….’, was it done under dry or moist conditions?

8.     Please provide a table of specimens names and abbreviations for better understanding of Figure 1-3, (there are some typing errors in Fig 1, please correct accordingly.

9.     Line 84, ‘surface roughness’ measurement which was mentioned in the ‘aim’ is missing, did you tested ? 

10.  Authors are advised to be consistent with their description particularly while describing materials, for example esthetic CAD/CAM or hybrid ceramics or CAD-CAM block etc. 

11.  Line 230, Figure 5 results and data and result description is missing. 

12.  Line 320-324, last paragraphs looks like a comment from someone else other than authors, please revise. 

13.  Conclusion section is very week, please summarise your results and key finding here.

Author Response

(The authors gave the same response as above.)

Reviewer 3 Report

Dear authors,

the article covers a very interesting topic and I support its publication.

I suggest some changes in order to improve the overall quality of the manuscript for the readers.

1) Line 14:

Please modify:

Twenty specimens of three esthetic CAD-CAM materials, Vita Enamic, Celtra® Duo, and Ceresmart, were prepared, the total number of specimens was 60. 

in:

Sixty specimens were prepared out of three esthetic CAD-CAM materials, Vita Enamic, Celtra® Duo, and Ceresmart (n=20).

2) Change also:

Randomly divided ten specimens of each material were subjected for the home and office bleaching 

in:

Specimens were randomly divided (n=10) according to the bleaching procedure (in office, home).

3) Line 42:

smoking, and food colorants such as red wine, tea, and coffee (2, 8, 9) 

A very recent and complete review about staining and color stability could be cited.

Please add the following reference:

Paolone G, Formiga S, De Palma F, Abbruzzese L, Chirico L, Scolavino S, Goracci C, Cantatore G, Vichi A. Color stability of resin-based composites: Staining procedures with liquids-A narrative review. J Esthet Restor Dent. 2022 Apr 9. doi: 10.1111/jerd.12912. Epub ahead of print. PMID: 35396818.

4) Line 122-3:

The formula not for ΔTP00 but for TP00, please remove Δ. The Δ in Translucency parameter is something that you can calculate between two readings at different intervals. 

5) Line 123-4:

The authors wrote:

“where where B is a black and W is a white background measured before and after the intervention, used to calculate ΔTP00 changes”

First of all: where is B and W in the formula?

When you place the correct formula (Lw - Lb, etc.) you should just write:

“where where B is a black and W is a white background used to calculate the TP”

6) Line 253-254:

Please cite again the abovr mentioned review.

7) Line 266:

what is: ΔT ??

8) Line 320-323:

Authors shall remove the following sentence:

“Authors should discuss the results and how they can be interpreted from the perspective of previous studies and of the working hypotheses. The findings and their implications should be discussed in the broadest context possible. Future research directions may also be highlighted.”

9) Line 327:

The authors wrote:

“Color recovery after red wine staining by using at-home and in-office bleaching appeared to be esthetic CAD-CAM material dependent”
I suggest to remove the word “esthetic”

Author Response

(The authors gave the same response as above.)

Round 2

Reviewer 2 Report

Thank you for making the suggested changes. The manuscript looks acceptable to me although it requires a few edits. Figures 1-3 are of low quality and unreadable, please provide high-resolution graphs for better understanding. Authors are encouraged to refine the discussion section mainly related to the staining difference between the materials used in this study i.e polymer-based ceramic and composite resin-base ceramics. 

Author Response

(The authors gave the same response as above.)

Reviewer 3 Report

The authors have addressed all the comments.

Author Response

Dear reviewer:

We thank the reviewer for your encouraging remark. Thank you very much for your comments that helped us improve this manuscript.